# Application of Magnesium and Calcium Sulfate on Growth and Physiology of Forage Crops under Long-Term Salinity Stress

**DOI:** 10.3390/plants11243576

**Published:** 2022-12-18

**Authors:** Khulan Sharavdorj, Ser-Oddamba Byambadorj, Yeongmi Jang, Jin-Woong Cho

**Affiliations:** 1Department of Crop Science, College of Agricultural and Life Sciences, Chungnam National University, Daejeon 34134, Republic of Korea; 2Laboratory of Forest Genetics and Ecophysiology, School of Engineering and Applied Sciences, National University of Mongolia, Ulaanbaatar 14201, Mongolia

**Keywords:** forage crop, salinity stress, magnesium sulfate, calcium sulfate, morphology, ion concentration, photosynthesis

## Abstract

Soil salinity is major threat to crop growth and reducing cultivated land areas and salt-resistant crops have been required to sustain agriculture in salinized areas. This original research was performed to determine the effectiveness of MgSO_4_ (MS) and CaSO_4_ (CS) for each species and assess changes in the physiology and growth of fodder crops after short and long-term salt stress. Six treatments (CON (control); NaCl (NaCl 100 mM); 1 MS (1 mM MgSO_4_ + 100 mM NaCl); 2 MS (2 mM MgSO_4_ + 100 mM NaCl); 7.5 CS (7.5 mM CaSO_4_ + 100 mM NaCl); and 10 CS (10 mM CaSO_4_ + 100 mM NaCl)) were applied to Red clover (*Trifolium pratense*) and Tall fescue (*Festuca arundinacea*) under greenhouse conditions. Cultivars were evaluated based on their dry weights, physiological parameters, forage quality, and ion concentrations. The biomass of both species decreased significantly under NaCl treatments and increased under the MS and CS treatments compared to solely salinity treatments. Salinity caused a decrease in the photosynthetic rate, but compared to CON, the MS and CS treatments yielded superior results. Moreover, the Na^+^/K^+^ ratio increased as Na^+^ concentration increased but crop quality (CP, NDF, ADF) did not show significant differences under salinity. Overall, we concluded that these *T. pratense* and *F. arundinacea* species demonstrated various responses to salinity, MS, and CS by different physiological and morphological parameters and it turned out to be efficient under salinity stress.

## 1. Introduction

Approximately 8% of the world’s irrigated land is located in arid and semi-arid areas where soil salinization occurs [1]. Watering is therefore required for high forage crop production in semi-arid and arid regions. Nonetheless, salinization is often related with irrigation due to soil salinity [2]. A saline soil is one with an electrical conductivity (EC) of the saturation extract (ECe) in the root zone that exceeds 4 dSm^−1^ (approximately 40 mM NaCl) at 25 °C and an exchangeable sodium of 15%. Most crop plants’ yields are reduced at this ECe, however many crops’ yields are reduced at lower ECe’s (i.e., kale yield was decreased under 6, 9 dS^−1^) [3,4]. Furthermore, salt stress influences the development of light-harvesting complexes and governs the state transition of photosynthesis [5]. Forage crop farming has historically been practiced because livestock is an essential source of income. Furthermore, forage expenses account for 50–70% of animal production operating input costs [6]. Forage crops can be fed to cattle directly or processed through partial drying feeds can be classified as bulky feeds or concentrates as a result of this processing. Bulky feeds, also known as fodder, are made from grass, cereal, and legume cropping, such as alfalfa, Lolium, or a combination of the two [7]. Harsh environmental conditions, salt, drought stresses, and high temperature during the growing season increase polysaccharides in cell walls and reduce soluble carbohydrates, thereby leading to increased acid detergent fiber (ADF) and neutral detergent fiber (NDF) [8], and decrease the crude protein (CP) [9]. It is believed that less essential elements absorbed by plants lead to increase fibers in forage [10]. Additionally, under medium salinity (50 mM), it is shown that nutritional elements, such as N, P, Mg^2+^, Na^+^, Cl, and S, and nutritional quality, such as the net energy and CP, and the relative feed value of forage slightly improved, but other essential nutrients, such as Ca^2+^, K^+^, were decreased [11]. Plant growth and developments are associated with the physiological responses that are associated with ion accumulation. Nutrients that are micro or macro are essentials to the plants adopt under salinity environment [12].

Calcium and magnesium are the most abundant cations in plants [13]. Plant nutrient deficiencies could occur when high concentrations of Na in the soil decrease the available Ca^2+^ and Mg^2+^. Calcium is a key macronutrient that protects plant development and survival in high saline environments [14]. The ion activities in the external solution have been linked to the effect of Ca^2+^ supplementation on salt-stressed roots. Root elongation is usually correlated with the activities of Na^2+^/Ca^2+^ in the external solution based on a simple ion exchange theory at the surface of the plasma membrane. Calcium sulfate (CaSO_4_) is often used as calcium fertilizer in agriculture sector [15]. Calcium sulfate is easily accessible to the plants due to the rich source of sulfur. It has a positive effect on the volume and quality of forage crop’s yield [16]. Magnesium is identified as an essential nutrient of plant [17]. Magnesium is essential elements to plants growth, and 75% of leaf Mg involved in biochemical processes, and 15%–20% of total Mg affiliated with chlorophyll pigments [18,19]. Moreover, even minor Mg deficiency can reduce biomass and plant sensitivity to external challenges by impairing biochemical and physiological processes [20]. Numerous studies have found that Mg is essential for adapting to a salt environment [21,22].

The ecological role of Fabaceae Is in nitrogen fixation with soil bacteria rhizobia [23]. This makes the Fabaceae family one of the important among plant kingdom and it is the third largest family in the world. Red Clover (*Trifolium pratense*) is a common legume species and it is widely used as forage crop, medicinal plant [24]. Red clover provides high-value feed for grazing livestock and, compared to other legumes, has a lower rate of protein degradation during ensiling [25]. Studies have proven that salinity reduced water potential in the leaves, affects the root length and decreased stem length, and biomass of clover [26,27]. Tall fescue (*Festuca arundinacea*) is a major cultivated cold-season perennial species, native to Europe. It is an extensively farmed forage grass around the world due to its high economic value, rapid growth, great reproductive capability, high nutritional content, and good palatability, and for soil conservation and land rehabilitation [28,29]. Under salinity stress, Tall fescue showed a lower density, poor growth (biomass decrease), and plant height reduction [28].

Although there have been several studies on the effects of salinity on plants, the majority of the studies have focused on crop seedlings with short-term salinity applications. Furthermore, the effect of MgSO_4_ and CaSO_4_ application on forage crops under long-term salt stress has not been thoroughly explored. As a result, the current study was designed to investigate and determine the effect of MgSO_4_, CaSO_4_ on forage crops grown in a saline environment. The study’s specific aims were to (a) determine the effectiveness of MgSO_4_ and CaSO_4_ for each species, and (b) assess the changes in physiology and growth of fodder crops after short and long-term salt stress during vegetative growth after overwintering in the greenhouse.

## 2. Results

### 2.1. Biomass

A variety of growth parameters (total biomass, aboveground biomass, root biomass, aboveground height, root length, and specific leaf area) have been calculated to determine the effect of CS and MS levels on the growth of *T. pretense* and *F. arundinacea* under salinity stress. The two-way ANOVA indicated a significant difference between species for all dependent variables except ABG biomass. There was a significant difference found in ABG height (*p* < 0.0001) and root length (*p* < 0.0001) between treatments. However, between species x treatments, a significant difference was found only ABG height (Table 1).

The heights and root lengths of *T. pratense* and *F. arundinacea* was decreased in NaCl treatments, and CaSO_4_ and MgSO_4_ enhanced the root lengths and heights (Figure 1). Moreover, 2 MS treatment found highest height among treatments (60 DAT, *p* = 0.0137; 120 DAT, *p* < 0.002) (Figure 1A) in *T. pratense* with both measurements, but *F. arundinacea* showed that 10 CS treatments were highest height at both measurements (60 DAT, *p* = 0.0412; 120 DAT, *p* < 0.002) (Figure 1B). However, throughout the experiment period, the control’s ABG height was higher than all of the treatments in both species, while root length was shorter during the 60 DAT and grew further during the 120 DAT in both species (Figure 1). Furthermore, root length most increased when 10 CS treatment was applied in *T. pratense* at 60 DAT, but during the next measurements (120 DAT), 1 MS treatments showed highest root length (60 DAT, *p* = 0.7879; 120 DAT, *p* = 0.4774) (Figure 1C). During the first measurement (60 DAT) for the *F. arundinacea*, root lengths were similar between treatments, with 10 CS treatments showing a slightly higher value when compared to the NaCl treatments, but 1 MS and 10 CS treatments showed increased root lengths at the second measurement (120 DAT), and it was similar between CON treatments (60 DAT, *p* = 0.4194; 120 DAT, *p* = 0.2166) (Figure 1D).

The biomass of above and belowground of both species was significantly affected by NaCl treatments (Figure 2). During the first (60 DAT) and second (120 DAT) measurements, the highest aboveground biomass of *T. pratense* was observed in CON and the lowest value was found in NaCl treatments, with MS treatments outperforming CS treatments (Figure 2A). By 60 DAT, 2 MS treatments had the highest aboveground biomass compared to the other treatments, however at 120 DAT, the maximum biomass was found in 10 CS except CON (Figure 2A). For the *F. arundinacea*’s, highest aboveground biomass was also found in CON and lowest was found in NaCl treatments and 10 CS treatments were higher among other treatments (Figure 2B). In case of belowground biomass of *T. pratense*, highest values were found in CON and the lowest value was found in NaCl treatments, and during the first measurement (60 DAT), MS treatments were slightly higher than CS treatments but after second measurement (120 DAT), belowground biomass of CS treatments were higher than MS treatments (Figure 2C). For the belowground biomass of *F.*, the highest values were found in 10 CS treatments and the lowest values were found in NaCl treatments, and CON was lower than CS, MS treatments. Furthermore, 2 MS found lowest values within the treatments, especially during the second measurement (120 DAT) it was similar with NaCl treatment (Figure 2D).

### 2.2. Specific Leaf Area (SLA) Measurements and SPAD

Specific leaf area (SLA) was calculated from leaf area index. SLA was found highest in CON during the first measurement in *T. pratense* and the lowest value occurred in 10 CS treatment. However, SLA during the second measurement (120 DAT), CON showed the lowest value and found highest value in 1 MS treatment (Figure 3A). For the *F. arundinacea*, the highest SLA value were found in CON and the lowest value was found in NaCl treatments throughout the experiment, and MS and CS treatments slightly increased the SLA (Figure 3B).

The comparison between treatments, NaCl treatment was significantly influenced the chlorophyll contents in *T. pratense*, as assessed by SPAD value. However, comparing between 60 DAT and 120 DAT, SPAD values were similar in *T. pratense* but NaCl influenced to the value of *F. arundinacea* at 60 DAT, but long-term salinity application (120 DAT) showed no significant decrease. For the MgSO_4_ and CaSO_4_ treatments, both treatments resulted in elevated SPAD values in both species comparing to the NaCl treatment throughout the experiment period, and 7.5 CS showed highest value in *T. pratense* at first measurement (60 DAT), and 10 CS at latest measurement (120 DAT) (Figure 3C). However, SPAD values were higher during the first measurement (60 DAT), and the highest value was found 1 MS and 2 MS at the second measurement (Figure 3D).

### 2.3. Photosynthesis Rate

When PAR levels increased in all species and treatments, the rate of photosynthesis increased (Figure 4). The results showed that the photosynthetic rate of *T. pratense* was higher under MS treatments than the CON, while values were lowest when treated with NaCl (Figure 4A,B). When plants were treated to CS treatments, the rate of *F. arundinacea*’s photosynthesis increased (Figure 4C,D). Throughout the trial, the application of salinity had a deleterious effect on both species. Furthermore, when compared to the CON, the MS and CS treatments yielded superior results.

### 2.4. Elemental Analysis

To demonstrate the effect of treatments on element distribution, we analyzed the concentrations of the ions Na^+^, P, K^+^, Ca^2+^, and Mg^2+^ in the stem, leaf, and root of *T. pratense* and *F. arundinacea*. Significant differences were found in the Na, Ca, K, Mg, and K content of the root, stem, and leaves, and were identified between treatments.

The result of *T. pratense* showed that Na concentration was increased in roots, stems, and leaves after salt treatments and found highest in the stem of NaCl, and Na concentration in roots where the highest in 10 CS and Na concentration in the leaf were lower than compared to the roots and stem, and lowest value was found leaf of CON. For the Ca concentration, highest value was found in stem of NaCl, and the lowest content was found in NaCl of root; other treatments were showed similar results. The K and Mg concentrations were found to be similar in all treatments, with larger concentrations in the stem and leaf, and lower concentrations in the roots. The highest concentrations were in the stem of 2 MS, the leaf of NaCl, and the lowest values were in the root of 10 CS, respectively. However, P concentrations were higher in the leaves than in the roots and stems, with the maximum concentration detected in the leaf of CON at 60 DAT. The elemental analysis results at 120 DAT revealed a 2-fold rise in Na in all treatments except CON, with the highest level found in the root of NaCl and the lowest value found in the stem of CON. However, Ca concentration was decreased in the stem and leaf in all treatments, but Ca concentration was increased in the roots, with the highest content found in the leaf of 10 CS and the lowest content found in the root of 1 MS. The amount of K was found to be lower in the stem of CON than in the other treatments, with the lowest content found in the root of NaCl and the highest value found in the stem of 2 MS. In general, Mg was decreased in stem of all treatments, but root and leaf were similar with previous measurements and between treatments highest content of Mg was found in leaf of 1 MS, and the lowest value was found in root of 1 MS. P content in root was sharply decreased in 120 DAT than 60 DAT measurements. However, it was similar between the treatments, and the P content tended to be higher in CS treatments and the highest content was found in root of 2 MS.

Furthermore, elemental analysis of *F. arundinacea* showed that similar trend of Na content with *T. pratense* species which was highest content was found in 10 CS treatment, but the leaf contained a higher content of Na in all treatments. For the Ca content, it was similar between treatments and the highest value was found in leaf of 7.5 CS, and the lowest value was found in root of 1 MS. K contents were found higher in the stem and leaf, and the highest content was found in the stem of CON, and the lowest value was found in the root of 10 CS. The salinity treatment decreased the K content in leaf (2-fold) than other treatments. Additionally, salinity decreased the content of Mg in the leaf (1.5-fold) compared other treatments. For the P content, it was similar between treatments and salinity also decreased the leaf’s P content around 1-fold, and the highest content was found in the stem of CON and the lowest value was found in the root of 10 CS. The result of long-term (120 DAT) treatment applications showed similar results with 60 DAT. The highest content of Na found in the stem of 10 CS and the lowest value was found in the root of CON. The Ca and contents were found higher in stems and leaves rather than roots, but K content was found higher in leaves than stems and roots. For the Mg and P contents, results showed that similar content between treatments and also salinity decreased the leaves’ Mg, P content about (0.5-fold) (Appendix A). Na^+^/K^+^ of *T. pratense* root has the highest ratio than the stem and leaves. Compared with 60 DAT, the Na^+^/K^+^ of roots with all treatments in 120 DAT increased by 5-fold in NaCl treatments (Figure 5A,B). For the ratio of Na^+^/K^+^ of *F. arundinacea*, the root and leaves ratio were the higher than the stem, but stem of 2 MS treatment had the highest ratio than the root at 60 DAT and it was decreased, and it was similar with the CON and ratio of stems were increased in 120 DAT except CON and 1 MS treatments (Figure 5C,D).

To identify the key parameters for assessing treatments effect in *T. pratense* and *F. arundinacea*, both morphological and macronutrients were used to plot heat map. As shown in Figure 6, the morphological and macronutrients of both species, grown under five treatments or CON, were used for hierarchical (row) clustering. This clear clustering demonstrates that, in comparison to CON conditions, salinity stress treatments alter morphological characteristics and macronutrients for both species. The heat map clearly reveals considerable variation in their morphological responses to salinity stress and CON (Figure 6).

Crop quality is determined by Crude protein, NDF, and ADF, contents which are shown in Table 2. Significance differences in CP, NDF, ADF content were identified between the treatments of *T. pratense* at 60 DAT. CP content in CON showed the lowest value and highest value was found in 7.5 CS and higher values of NDF, ADF were found in CON, and MS treatments. Moreover, there was not significant difference in CP found between treatments at 120 DAT. The minimum CP content was found in 2 MS followed by NaCl, and the highest content was found in 1 MS. In addition, higher content of NDF, ADF were found in CON, 7.5 CS treatments.

The significant difference was observed between the treatments in the CP and NDF contents of *F. arundinacea* at both measurements. The CP content of CON, 10 CS, and 7.5 CS showed lower contents and higher contents than were found in NaCl and 7.5 CS treatments at 60 DAT and 120 DAT, respectively. For the NDF and ADF, higher contents were found in 1 MS, followed by NaCl and lowest values were found in 10 CS at 60 DAT. The latest measurement showed the lowest content of NDF and ADF were found in 1 MS and NaCl, and the highest values were found in 10 CS and CON, respectively.

## 3. Discussion

Salinity is more common in semi-arid regions of the world, and shifting weather patterns have increased the frequency of droughts [30]. Crops are sensitive to salinity, and NaCl is toxic when it accumulates in the roots [31,32,33]. Salinity stress reduces plant physiological and morphological processes by causing osmotic stress, and inducing ionic and nutrient imbalance, resulting in decreased crop yield [34,35]. The current study examined the effects of various magnesium and calcium sulfate concentrations on two types of forage crops that were subjected to long-term salt stress. The study proposed that MgSO_4_ and CaSO_4_ improve crop salt tolerance and increase biomass at short-term (60 DAT) and long-term (120 DAT) treatments.

The current study found that plant growth (total height, stem, and root lengths) was reduced in all treatments compared to the CON, and the height reduction in NaCl was highly pronounced when compared to other treatments, although the application of MgSO_4_ and CaSO_4_ showed superior adaptability when compared to solely salt applied plants (Figure 1). Previously conducted investigations on other plants complement our findings [34,36]. Furthermore, the aboveground and belowground biomass was severely reduced in both species under salinity, while extra application of MgSO_4_ was most effective in increasing biomass in *T. pratense*, and CaSO_4_ was most effective in increasing biomass in *F. arundinacea* throughout the experiment (Figure 2).

Previous research on various plants found that NaCl stress reduced aboveground and belowground biomass [37,38,39,40]. The loss in biomass indicates the plant’s stress response, which is governed by growth inhibition in response to increased salinity [41]. Salinity generates an ionic imbalance, which disrupts K^+^ uptake by the root, and excess Na^+^ promotes ROS, resulting in increased oxidative stress [42]. Crop salt tolerance declines beyond 2 dSm^−1^, and when salinity levels reach this level, crop development and output suffer [30]. A previous study suggested that NaCl and its mixture (CaSO_4_ + NaCl, MgSO_4_ + NaCl) significantly increased the seedling height and biomass, and germination was greatly reduced under salinity for *T. pratense* and *F. arundinacea* [43,44]. Furthermore, salinity decreased the SLA and SPAD values, and the SLA result was similar between the short-term and long-term treatments in both species (Figure 3). Some studies agreed to our finding that leaf area and SPAD values were reduced by increasing salinity concentration [45,46,47]. The reduction in leaf area under environmental stress, could be explained by an adaptive morphological strategy that of limiting water loss through transpiration and slowing of cell growth of young leaves [48]. In a long-term salinity, the SPAD value did not decrease in *F. arundinacea*, which could be explained that salt tolerance plants have increased or remained constant in chlorophyll content as a biochemical measure of salt tolerance in plants [49]. Furthermore, photosynthesis is the most crucial process that occurs in plants. Our findings demonstrated that the use of MS and CS treatments was able to overcome the inhibitory effect of salt stress on the photosynthetic rate (Figure 4). Salinity has both direct and indirect effects on photosynthetic productivity, and one method to solve those effects is to modify the activity and expression levels of enzymes involved in chlorophyll production and photosynthesis, while another method is to regulate pathways, such as antioxidant enzyme systems [50,51]. In addition, calcium sulfate improved growth and physiology of plants that were exposed to the salinity [36] and calcium adjusts the chloroplast NAD^+^ kinase activity and regulate the activity of the phosphatase enzymes in the carbon reduction cycle [52,53]. Magnesium has a main role in photosynthesis, and is associated with the process in the chloroplast, and it is required for the synthesis of chlorophyll [54,55]. According to the Hauer-Jákli and Tränkner [55], plants response to the Mg deficiency is to reduce the photosynthesis rate and increase the leaf area, increasing the carbohydrate content of the shoots that lead to the higher photosynthesis performance [56], and this supports our results that Mg application increased the photosynthesis rate in both species throughout the experiment compared to the control.

Appendix A summarizes Na, Ca, K, Mg, and P concentrations in the different part of the *T. pratense* and *F. arundinacea*. Salinity stress caused an increase in levels of Na in all parts of the *T. pratense* and *F. arundinacea* in all treatments. The highest Na concentration was found in the roots followed by stem. High amount of Na^+^ in cells damages membrane severely, resulting growth reduction and death of plant [57]. Ca concentration in the leaf was found highest in *T. pratense* in short-term salinity exposure, which could be explained by the rapid increase in cytosolic calcium concentration [58]. The concentration of K and Mg found highest in stems followed by leaf in both species and the salinity decreased the K and Mg contents. According to the [57], high uptake of Na^+^ over K^+^ ions in plant cause inconsistent Na^+^/K^+^ in the roots under salinity stress, and it could lead to the low rate of growth and decreased biomass. The changes in Na^+^/K^+^ can represent the degree of plant damage and the change in nutrient balance in plants. Phosphorus (P) is an essential macro-element that involved many metabolic processes, such as photosynthesis and respiration, and P deficiency is one of major abiotic factor that limits plant growth and yield of crop [59,60]. The P concentration in the leaf decreased under salinity, and generally, the root of *F. arundinacea* contained lowest P concentration during the experiment, but *T. pratense*’s stem contained the lowest P concentration, and, when comparing the two species, *T. pratense* had higher P concentration than *F. arundinacea*. Generally, legume species are considered to be rich in phosphorus than perennial grasses [61]. Our previous study [44] also showed similar results that legume species showed higher P concentrations. A heatmap is a visual method that can be used to investigate complex relation between multiple parameters collected from different treatments and its often functional combine heatmap with hierarchical clustering, which is a way of assembling items in hierarchy based on the distance or similarity between treatments [62]. After salinity stress, most of the measured traits in *T. pratense* approached to CON and 10 CS in *F. arundinacea* (Figure 6).

Crop quality is often indicated by crude protein and fiber [63]. The effects of salt stress on the protein in plants vary based on the environment and cultivar [64]. Our results suggest that treatments including sole salinity decreased the NDF and ADF content and increased the CP contents in both species. According to the Hu et al. [63], salinity decreases the photo assimilates and that directly leading to a reduction in NDF and ADF content. Moreover, fiber synthase activity inhibited under abiotic stresses [65]. A previous study reported the opposite results to our finding that higher salinity increased the NDF and decreased the CP content [66]. Our results showed a significant decrease in growth variables, dry weight, SLA, and photosynthesis rate; these are the most relevant parameters to evaluate the inhibition of growth induced under salinity stress. However, crop quality showed improved results under treatments. Therefore, it could indicated salt tolerance in both species.

## 4. Materials and Methods

### 4.1. Plant Materials and Experimental Design

The experiments were carried out in a greenhouse at Chungnam National University in Daejeon, South Korea, from November 2021 to October 2022. Seeds of *Trifolium pratense* L. and *Festuca arundinacea* Shreb. were provided by the National Institution of Animal Science of Korea. Seeds of *T. pratense* (2 kg/ha) and *F. arundinacea* (1 kg/ha) were sown in 38.5 L pots (55 × 35 × 20 cm) with sandy clay soil on 4, November 2021. Soils were well mixed with 700 kg/ha CaCO_3_, 15 kg/ha NO_3_, and manure, and soil pH were 7.3. Seedlings emerged after 3 to 10 days depending on species, and seedlings were irrigated every three days until December, then once a week throughout the winter (December to March).

### 4.2. Experimental Treatments

Treatments started on March 10 for the *T. pratense* and June 15 (due to low growth rate) to the *F. arundinacea*. Experimental treatments were: control—namely CON (no stress, no additional application of CaSO_4_, and MgSO_4_); salinity stress—namely NaCl (100 mM NaCl); salinity + MgSO_4_—namely 1 MS (MgSO_4_ 1 mM + NaCl 100 mM); 2 MS (MgSO_4_ 2 mM + NaCl 100 mM); and salinity + CaSO_4_—namely 7.5 CS (CaSO_4_ 7.5 mM + NaCl 100 mM); 10 CS (CaSO_4_ 10 mM + NaCl 100 mM). Until the completion of the experiment, solutions were applied once every four days. Each treatment had three replicates (three pots), and each pot was entirely filled with a solution of the appropriate concentration before being completely drained from the bottom.

Soil samples were collected from each treatment and measured the pH, electronic conductivity and salt by PC8500 Portable pH/Conductivity Meter Kit [67].

Table 3 shows soil pH, EC, and salt content of *T. pratense* and *F. arundinacea*, with six treatments at 60 DAT (60 days after treatments) and 120 DAT (120 days after treatments). The application of NaCl, MS, and CS were increased the soil pH, EC, and salt content compared to control.

### 4.3. Measurement of Biomass

The first measurement was done on 60 DAT, and second measurement was done on 120 DAT, after each measurements plant’s aboveground parts were harvested, and experiments finished October 2022. Four plants from three pots each treatment were carefully harvested and thoroughly washed with tap water.

### 4.4. Measurements of Specific Leaf Area (SLA) and SPAD

The color of the leaves was measured using a chlorophyll meter (SPAD) in the morning and expressed as a SPAD value (SPAD-502Plus, Konica Minolta Sensing Americas, Inc, Ramsey, NJ, USA). Leaf area was measured with a leaf area meter (LI-3100, Li-Cor Biosciences, Lincoln, NE, USA) right after it was collected from the pot. After separating the leaves, stems, and roots, and after measurements, plant parts were oven-dried at 80 °C to calculate biomass. Specific leaf area (*SLA*) was calculated as the ratio of leaf area to leaf dry mass (m^2^ kg^−1^) [68].
(1)SLA=A/ML
where *A* is the area of a given leaf or all leaves of a plant, and *ML* is the dry mass of those leaves.

### 4.5. Measurement of Photosynthesis Rate

Net photosynthetic rate was measured between 08:00 and 11:00 using a portable open-flow gas-exchange system (LI-6400, Li-Cor Biosciences, Lincoln, NE, USA). Light intensity was 0, 300, 500, 1000, and 1500 (PPFD) μmol m^−2^ s^−1^, leaf temperature to 30 °C, the leaf-air vapor pressure difference to approximately 1.5 kPa, and the ambient CO_2_ concentration to 400 μmol^−1^ during measurements.

### 4.6. Crude protein, ADF, NDF and Elemental Analysis in Plant Tissues

Elemental analysis was measured at the soil environment analysis center at Chungnam National University, following the nitric acid digestion method. Dried samples were ground into a fine powder, and 2 g of dry sample pulverized to about 40 mesh, and liquid is heated gradually to a low temperature until all the red gas of NO_2_ is blown off the hot plate, then it is gradually heated to a high temperature for about 1 h to obtain a clear decomposition solution. After cooling the decomposition solution, we added HCl and about 50 mL of distilled water, then slightly heated it again. After cooling for a while, we used No. 6 filter paper and added distilled water to adjust the filtrate to 250 mL. The resulting solutions were analyzed for Ca, Na, Mg, K, and P by using Inductively Coupled Plasma, Optical Emission Spectrometer (ICP-OES, Thermo iCAP6400 Series). The percentage of CP was determined by using Kjeldahl equipment following the methods of the Association of Official Analytical Chemists (AOAC) [69]. Neutral detergent fiber (NDF) and acid detergent fiber (ADF) were analyzed following the Van Soest methodology [70].

### 4.7. Statistical Analysis

All the statistical analyzes were performed using the Statistical Analysis Software (SAS) package [71]. Data analysis included two-way ANOVA (species and treatments), and Duncan’s test was used to estimate the least significant range between means (*p* < 0.01). All measurements represent the means and standard errors (SE).

## 5. Conclusions

Climate change leads to soil salinization and the industrialization negatively affects freshwater that leads to a decrease in crop production. This study evaluated the physiological comparisons of *T. pratense* and *F. arundinacea* under NaCl and Magnesium sulfate and Calcium sulfate. However, each species responded differently to different treatments, but NaCl stress drastically decreased plant growth and caused ion imbalance for both species. MgSO_4_ applied to plants of *T. pratense* and CaSO_4_ applied to plants of *F. arundinacea* performed better when compared to sole salt treatment, as shown through a lesser reduction in plant growth, maintained chlorophyll content, and improved photosynthetic rate. Therefore, identified plant materials can be a source for deeper insight into determining effective way to improve forage crop’s growth and yield under salinity environment.

## Figures and Tables

**Figure 1 plants-11-03576-f001:**
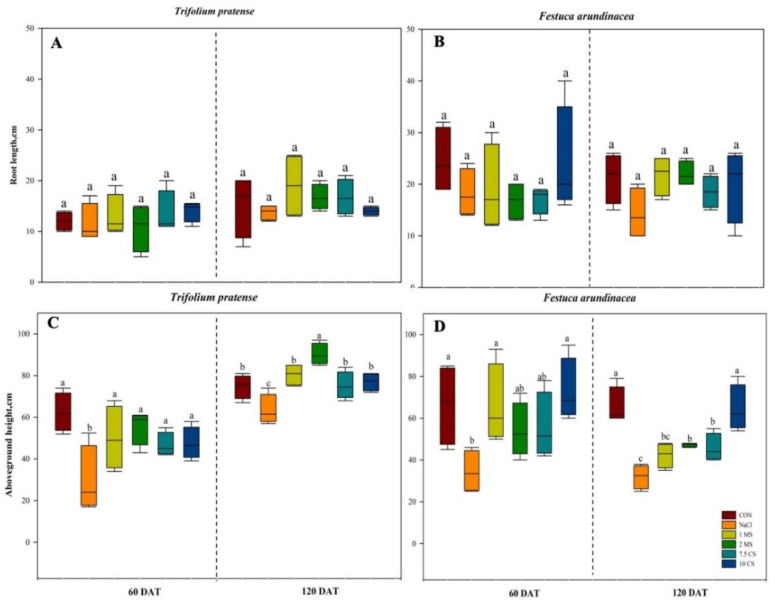
Evaluation of *T. pratense* (**A**,**C**) and *F. arundinacea* (**B**,**D**) by analyzing height and root length under six different treatments. Plants were treated with 100 mM NaCl solution; 1 mM MgSO_4_ + 100 mM NaCl (1 MS); 2 mM MgSO_4_ + 100 mM NaCl (2 MS); 7.5 mM CaSO_4_ + 100 mM NaCl (7.5 CS); and 10 mM CaSO_4_ + 100 mM NaCl (10 CS), within calculated 60 DAT and 120 DAT. Vertical bars represent standard errors (*N* = 4). Different lowercase letters indicate significant differences among the treatments at α = 0.05.

**Figure 2 plants-11-03576-f002:**
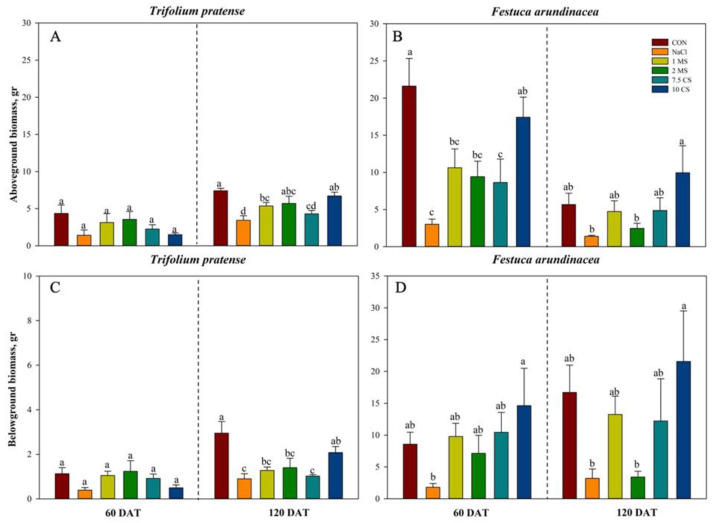
Evaluation of *T. pratense* (**A**,**C**) and *F. arundinacea* (**B**,**D**) by analyzing aboveground biomass and root biomass under six different treatments. Plants were treated with 100 mM NaCl solution; 1 mM MgSO_4_ + 100 mM NaCl (1 MS); 2 mM MgSO_4_ + 100 mM NaCl (2 MS); 7.5 mM CaSO_4_ + 100 mM NaCl (7.5 CS); 10 mM CaSO_4_ + 100 mM NaCl (10 CS), within calculated 60 DAT, and 120 DAT. Vertical bars represent standard errors (*N* = 4). Different lowercase letters indicate significant differences among the treatments at α = 0.05.

**Figure 3 plants-11-03576-f003:**
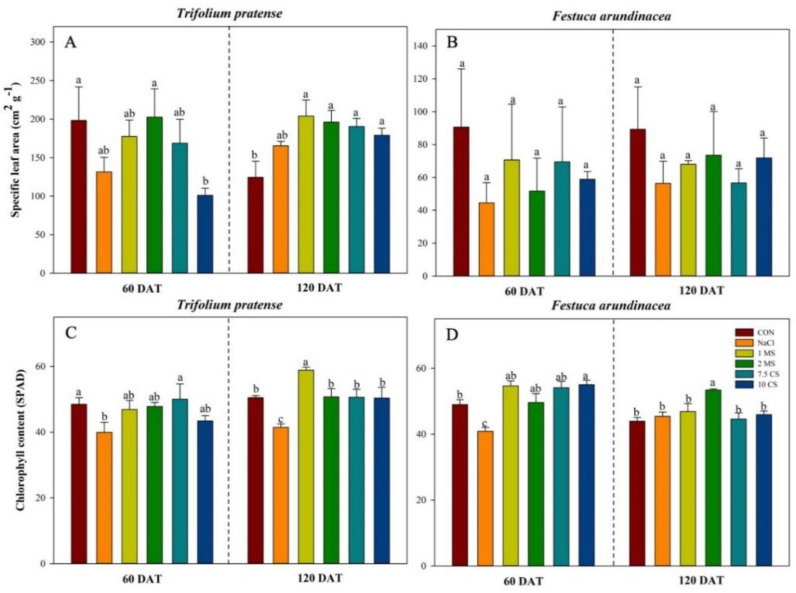
Course of SPAD value, and specific leaf area of *T. pratense* (**A**,**C**) and *F. arundinacea* (**B**,**D**) in 60 DAT and 120 DAT. Relative chlorophyll concentration assessed by SPAD measurements. Measurements were made on two species of each of the four biological replicates per treatment; four individual SPAD values per leaf were averaged. Vertical bars represent standard error (*N* = 4). Different lowercase letters indicate significant differences among the at α = 0.05.

**Figure 4 plants-11-03576-f004:**
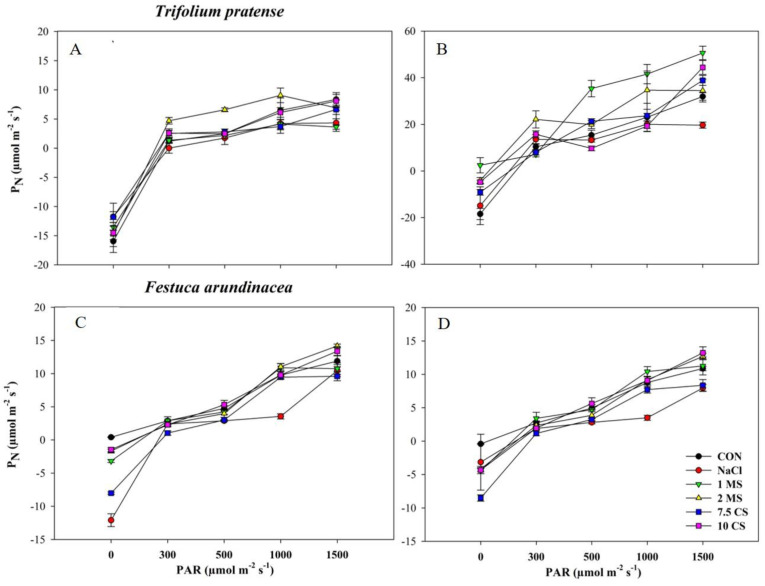
Photosynthetic rates of six different treatments under 400 μmol·mol^−1^ CO_2_ at the PAR value from 0 to 1500 µmol·m^−2^ s^−1^ with the air ambient temperature. (**A**) 60 DAT; (**B**) 120 DAT in *T. pratense*; (**C**) 60 DAT; (**D**) 120 DAT in *F. arundinacea*. Vertical bars represent standard errors (*N* = 4).

**Figure 5 plants-11-03576-f005:**
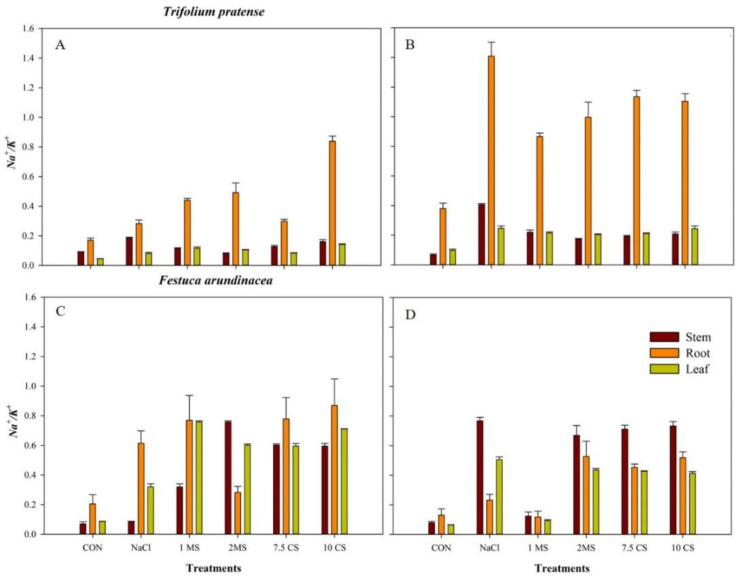
Na^+^/K^+^ ratio of *T. pratense* 60 DAT (**A**), 120 DAT (**B**) and *F. arundinacea* 60 DAT (**C**), 120 DAT (**D**) stem, root, and leaves of under different treatments.

**Figure 6 plants-11-03576-f006:**
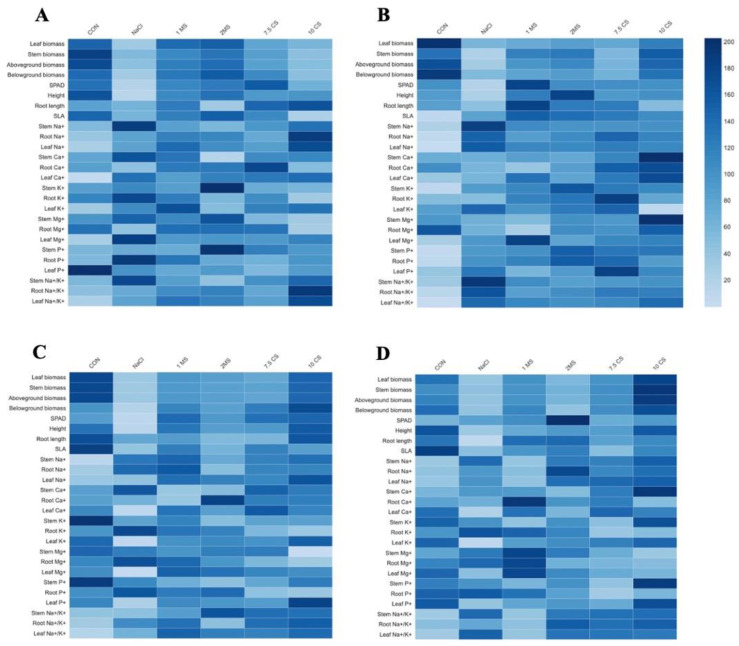
Heat map for morphological parameters and macronutrients under different treatments in *T. pratense* at (**A**) 60 DAT; (**B**) 120 DAT, and *F. arundinacea* (**C**) 60 DAT; (**D**) 120 DAT.

**Table 1 plants-11-03576-t001:** Two-Way ANOVA between species and treatments.

Measurements	Species			Treatments			Species × Treatments		
	D	F	*p* Value	D	F	*p* Value	D	F	*p* Value
Total Biomass	1	15.32	<0.0004	5	0.81	0.4948	5	0.37	0.9457
Aboveground biomass	1	0.61	0.4386	5	3.92	0.0061	5	2.21	0.0746
Belowground biomass	1	26.85	<0.0001	5	2.67	0.0377	5	2.03	0.0977
Height	1	206.03	<0.0001	5	14.16	<0.0001	5	8.58	<0.0001
Root length	1	12.88	0.0010	5	7.42	<0.0001	5	5.60	0.0006
Specific leaf area	1	169.44	<0.0001	5	1.23	0.3168	5	3.48	0.0115
SPAD	1	124.93	<0.0001	5	2.86	0.0435	5	3.24	0.0027

**Table 2 plants-11-03576-t002:** Crop quality indicators (%).

Species	Date		Treatments					
			CON	NaCl	1 MS	2 MS	7.5 CS	10 CS
*T. pratense*	60 DAT	CP	15.61 ^c^	16.96 ^b^	16.64 ^bc^	16.01 ^bc^	18.22 ^a^	15.84 ^c^
		NDF	36.27 ^a^	30.31 ^b^	31.1 ^b^	34.39 ^a^	32.33 ^b^	31.1 ^b^
		ADF	28.71 ^a^	20.85 ^c^	23.51 ^b^	22.31 ^b^	20.65 ^c^	18.05 ^d^
	120 DAT	CP	15.22 ^a^	15.09 ^ab^	15.83 ^a^	14.42 ^b^	15.74 ^a^	15.14 ^ab^
		NDF	44.03 ^a^	40.6 ^b^	40.51 ^b^	35.93 ^c^	43.57 ^a^	42.24 ^ab^
		ADF	36.63 ^a^	29.17 ^c^	31.98 ^b^	25.64 ^d^	32.25 ^b^	29.49 ^c^
*F. arundinacea*	60 DAT	CP	17.6 ^c^	20.38 ^a^	19.13 ^b^	18.45 ^bc^	19.13 ^b^	17.84 ^c^
		NDF	62.62 ^bc^	65.68 ^ab^	67.39 ^a^	59.07 ^c^	60.08 ^c^	50.9 ^d^
		ADF	28.45 ^a^	27.23 ^ab^	28.36 ^a^	26.21 ^b^	26.63 ^ab^	26.01 ^b^
	120 DAT	CP	12.37 ^c^	15.1 ^a^	14.01 ^b^	13.64 ^b^	12.5 ^c^	15.46 ^a^
		NDF	61.37 ^bc^	61.82 ^bc^	58.6 ^c^	60.98 ^bc^	64.15 ^b^	67.93 ^a^
		ADF	29.53 ^a^	27.33 ^b^	29.01 ^ab^	27.65 ^ab^	27.88 ^ab^	28.91 ^ab^

Different lowercase letters indicate significant differences across the treatments at α = 0.05.

**Table 3 plants-11-03576-t003:** Soil pH, electrical conductivity (EC), and salt values with six different treatments.

Species	Date	Treatments	pH	EC (ds/m)	Salt (ppt)
*T. pratense*	60 DAT	CON	7.36	0.036	0.02
		NaCl	8.46	0.156	0.7
		1 MS	8.02	0.224	0.11
		2 MS	8.24	0.193	0.09
		7.5 CS	8.44	0.105	0.05
		10 CS	7.65	0.46	0.21
	120 DAT	CON	7.9	0.024	0.01
		NaCl	8.5	0.929	0.44
		1 MS	7.52	0.559	0.26
		2 MS	7.9	0.441	0.2
		7.5 CS	7.5	0.291	0.13
		10 CS	7.65	0.522	0.15
*F. arundinacea*	60 DAT	CON	5.28	0.029	0.04
		NaCl	8.37	0.211	0.1
		1 MS	7.7	0.181	0.09
		2 MS	7.73	0.173	0.05
		7.5 CS	7.66	0.29	0.15
		10 CS	8.06	0.104	0.05
	120 DAT	CON	5.79	0.025	0.05
		NaCl	7.34	0.057	0.27
		1 MS	7.29	0.03	0.02
		2 MS	7.57	0.064	0.29
		7.5 CS	7.22	0.061	0.29
		10 CS	8.14	0.19	0.09

## Data Availability

The data used is primarily reflected in the article. Other relevant data is available from the authors upon request.

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
