# Peer review of "Application of Magnesium and Calcium Sulfate on Growth and Physiology of Forage Crops under Long-Term Salinity Stress"

_plants, 2022, doi:10.3390/plants11243576_

Round 1
Reviewer 1 Report
The manuscript is of good quality and has significant contribution on both scientific and practical levels.
I recommend the publication of the manuscript.
Author Response
Dear the Editor and Reviewers
On behalf of my co-author, I thank the editor and reviewers for the valuable comments on the manuscript entitled “Application of Magnesium and Calcium Sulfate on Growth and Physiology of Forage Crops Under Long-Term Salinity Stress”. We have tried to address all the editor’s and reviewers’ concerns in a proper way and believe that the paper has improved considerably.
I would be happy to make further corrections if necessary and look forward to hearing from you soon.
Sincerely,
Jin-Woong Cho
Professor
Chungnam National University
E-mail: [email protected]
Reviewer 2 Report
In this article, author investigated the impact of the “Application of Magnesium and Calcium Sulfate on Growth and Physiology of Forage Crops under Long-Term Salinity Stress”
This is an interesting paper, and it brings some interesting and useful information, and I think that the manuscript has a potential to be published in the journal. However, there are some vague cases in the paper. The below revisions is necessary:
· The writing quality is quite poor, the ideas is not that easy to follow, authors should send the manuscript to English editing services in order to improve grammar/syntax and readability.
· Abbreviation provided in line 12 and 13 is confusing please simplify.
· What MS CP, NDF, ADF and CS stands for in line 18 and 21? Please clarify it at least once in the Abstract.
· Line 84-85: The study's specific aims needs to be provided in more details.
· More specification is needed in M&M section, i.e.,
o The amount of seeds used pot-1
o Irrigation water applied
o The experimental layout
o The time of performing Specific leaf area (SLA) and SPAD as well as photosynthesis rate
· Figure 3, C and D is not clear enough to understand, I would rather recommend to be produced in a column shape.
· More recent publications should be added to your discussion related to your work
Author Response

(The authors gave the same response as above.)

Reviewer 3 Report
Paper Title- Application of magnesium and calcium sulfate on growth and physiology of forage crops under long-term salinity stress
Ms ID/ Ref no.:
Journal Name: Plants
Authors: Khulan Sharavdorj, Ser-Oddamba Byambadorj, Yeongmi Jang and Jin-Woong Cho
Reviewer Recommendation: Minor Revision
General comments-
1. Grammatical errors are present, please revise the whole manuscript to remove any possible grammatical and typos errors.
2. Error in sentence formation, please revise the whole manuscript to avoid the use of long sentences. Paraphrasing requires at several places in the Ms.
3. The first use of abbreviation should be accompanied by its full form and thereafter, the abbreviated form must be used throughout the Ms. Sentence should not begin with abbreviation.
4. Please check the sequence of different sections. Ideally it must be Introduction, Materials and Methods, Results, Discussion, Conclusion.
5. Please use meta images for the figures as in the current Ms. the figures become hazy upon zoom in.
6. There are lots of statements/sentences and scientific facts/hypothesis throughout the manuscript that need proper validation with previous studies, especially from recently published work that are largely missing in the present manuscript. Therefore, some recommended papers listed below should be cited in this Ms.
· El-Ezz, S.F.A., Al-Harbi, N.A., Al-Qahtani, S.M., Allam, H.M., Abdein, M.A. and Abdelgawad, Z.A., 2022. A comparison of the effects of several foliar forms of magnesium fertilization on ‘superior seedless’(Vitis vinifera L.) in saline soils. Coatings, 12(2), 201.
· Gupta, P. and Seth, C.S., 2020. Interactive role of exogenous 24 Epibrassinolide and endogenous NO in Brassica juncea L. under salinity stress: Evidence for NR-dependent NO biosynthesis. Nitric Oxide, 97, 33-47.
· Irakoze, W., Quinet, M., Prodjinoto, H., Rufyikiri, G., Nijimbere, S. and Lutts, S., 2022. Differential effects of sulfate and chloride salinities on rice (Oryza sativa L.) gene expression patterns: A comparative transcriptomic and physiological approach. Current Plant Biology, 29, 100237.
· Swaefy, H.M. and Abdallh, A.M., 2021. Mitigation of salinity stress in Fenugreek plants using zinc oxide nanoparticles and zinc sulfate. Journal of Plant Production, 12(12), 1367-1374.
· Prajapati, P., Gupta, P., Kharwar, R.N. and Seth, C.S., 2022. Nitric oxide mediated regulation of ascorbate-glutathione pathway alleviates mitotic aberrations and DNA damage in Allium cepa L. under salinity stress. International Journal of Phytoremediation, 1-12.
· Kafi, M., Nabati, J., Ahmadi-Lahijani, M.J. and Oskoueian, A., 2021. Silicon compounds and potassium sulfate improve salinity tolerance of potato plants through instigating the defense mechanisms, cell membrane stability, and accumulation of osmolytes. Communications in Soil Science and Plant Analysis, 52(8), 843-858.
· Kucukyumuk, Z. and Suarez, D.L., 2021. The effect of selenium on salinity stress and selenate–sulfate comparision in kale. Journal of Plant Nutrition, 44(20), 2996-3004.
· Agnihotri, A. and Seth, C.S., 2016. Exogenously applied nitrate improves the photosynthetic performance and nitrogen metabolism in tomato (Solanum lycopersicum L. cv Pusa Rohini) under arsenic (V) toxicity. Physiology and Molecular Biology of Plants, 22(3), 341-349.
· Hassan, A., Amjad, S.F., Saleem, M.H., Yasmin, H., Imran, M., Riaz, M., Ali, Q., Joyia, F.A., Ahmed, S., Ali, S. and Alsahli, A.A., 2021. Foliar application of ascorbic acid enhances salinity stress tolerance in barley (Hordeum vulgare L.) through modulation of morpho-physio-biochemical attributes, ions uptake, osmo-protectants and stress response genes expression. Saudi Journal of Biological Sciences, 28(8), 4276-4290.
Abstract
1. The content of the Abstract section does not provide brief introduction of the experiment, and the major implications or the future aspects of the study can be added.
Introduction
1. Line 34-35, please give the example of decrease in yield of the crop at different electrical conductivity levels.
2. Line 45, please write full form of “ADF and NDF”.
3. Line 47, please specify what do you mean by “medium salinity”?
4. Line 69-77, please also give an introduction on how the crop productivity of red clover and tall fescue is affected with the salinization of the soil.
Results
1. Line 92, please use the abbreviated from of calcium and magnesium sulfate to avoid the increase in word count.
2. Please justify why the different P-values have been considered for different measurements in the Table 1.
3. Line 106-119, please specify wherever the results depict a significant difference.
Materials and Method
1. Please mention the ratio in which CaCO3, and NO3, were mixed in the soil.
2. Please use symbol “+” rather than writing as plus.
3. Please write the full form of DAT for the first time when used, and afterwards switch to the usage of abbreviated form.
4. Please specify the number of seedlings that were used to study for each pot after final thinning for a better understanding of number of plants harvested at 120 DAT.
Discussion
1. Please provide a brief validatory line for the findings of the papers rather than providing a direct reference.
2. The discussion for the individual effect that is for 60 DAT and 120 DAT is well written. However, the discussion for the shift in trends from 60 DAT to 120 DAT needs to be strengthened with proper validation.
Conclusions
1. Please rewrite the conclusion directed towards the research impact and the findings of the paper in
References
1. Please check the formatting and editing according to the journal’s requirement.
Table 1
1. Please provide the concentration of GR24 for which the data is provided.
Figure 1
1. Please rewrite the legend in detail as “height” can be confused with the plant height or shoot height.
2. Please check and confirm the number of biological replicates considered for study are three or four. Kindly cross check with the material and methods section as well and in all the figures of the Ms.
Figure 3
1. Please specify the 60 DAT and 120 DAT in the figure 3A, and B.

Author Response

(The authors gave the same response as above.)
